# Characterization and Molecular Modelling of Non-Antibiotic Nanohybrids for Wound Healing Purposes

**DOI:** 10.3390/pharmaceutics15041140

**Published:** 2023-04-04

**Authors:** Caterina Valentino, Tomás Martínez Rodríguez, Ana Borrego-Sánchez, Pablo Hernández Benavides, Francisco Arrebola Vargas, José Manuel Paredes, Silvia Rossi, Claro Ignacio Sainz Díaz, Giuseppina Sandri, Pietro Grisoli, María del Mar Medina Pérez, Carola Aguzzi

**Affiliations:** 1Department of Drug Sciences, University of Pavia, Viale Taramelli 12, 27100 Pavia, Italy; caterina.valentino01@universitadipavia.it (C.V.); silvia.rossi@unipv.it (S.R.); g.sandri@unipv.it (G.S.); pietro.grisoli@unipv.it (P.G.); 2Department of Pharmacy and Pharmaceutical Technology, Cartuja Campus, University of Granada, 18071 Granada, Spain; pabloj@ugr.es (P.H.B.); mdelmar@ugr.es (M.d.M.M.P.); carola@ugr.es (C.A.); 3Instituto de Ciencia Molecular, Universitat de València, Carrer del Catedrátic José Beltrán Martinez 2, 46980 Paterna, Spain; 4Department of Histology, Institute of Neurosciences, Centre for Biomedical Research (CIBM), University of Granada, 18071 Granada, Spain; fav@ugr.es; 5Nanoscopy-UGR Laboratory, Department of Physical Chemistry, Unidad de Excelencia en Quimica Aplicada a Biomedicina y Medioambiente UEQ, University of Granada, Cartuja Campus, 18071 Granada, Spain; jmparedes@ugr.es; 6Instituto Andaluz de Ciencias de la Tierra (CSIC-University of Granada), 18100 Armilla, Spain; ci.sainz@csic.es

**Keywords:** bentonite, halloysite, chlorhexidine, spray drying, molecular modelling, wound healing, chronic wounds, biocompatibility, antimicrobial properties, antibiotic resistance

## Abstract

The healing process of chronic wounds continues to be a current clinical challenge, worsened by the risk of microbial infections and bacterial resistance to the most frequent antibiotics. In this work, non-antibiotic nanohybrids based on chlorhexidine dihydrochloride and clay minerals have been developed in order to design advanced therapeutic systems aimed to enhance wound healing in chronic lesions. To prepare the nanohybrids, two methodologies have been compared: the intercalation solution procedure and the spray-drying technique, the latter as a one-step process able to reduce preparation times. Nanohybrids were then fully studied by solid state characterization techniques. Computational calculations were also performed to assess the interactions between the drug and the clays at the molecular level. In vitro human fibroblast biocompatibility and antimicrobial activity against *Staphylococcus aureus* and *Pseudomonas aeruginosa* were assessed to check biocompatibility and potential microbicidal effects of the obtained nanomaterials. The results demonstrated the effective organic/inorganic character of the nanohybrids with homogeneous drug distribution into the clayey structures, which had been confirmed by classical mechanics calculations. Good biocompatibility and microbicidal effects were also observed, especially for the spray-dried nanohybrids. It was suggested that it could be due to a greater contact area with target cells and bacterial suspensions.

## 1. Introduction

In healthy individuals, wound healing takes place through a cascade of partially overlapping physiological phases, which require the presence of growth factors, cytokines, fibroblasts, endothelial and immune cells, to cause damaged tissue to repair itself [1]. However, when the cascade fails (especially in the case of chronic diseases or severe burns), non-healing wounds can be formed, which experience chronic inflammation. Chronic wounds are very susceptible to infection because the necrotic tissue in the wound site promotes the formation of complex bacterial and fungal biofilms [2], which are extremely resistant to antibiotic treatments. The persistence of biofilms increases tissue damage, as well as morbidity and mortality rates in patients, and leads to a significant socio-economic impact on healthcare costs worldwide. Moreover, systemic and local long-term administration and misuse of antibiotics in infected wounds have markedly increased antibiotic resistance [3].

Proper therapeutic interventions, especially those which are based on non-antibiotic approaches, such as the use of antiseptic compounds, bacteriophages, negative pressure and hyperbaric oxygen therapy, therefore become necessary to restore skin functionality in chronic wounds [4]. Among antiseptics, chlorhexidine is a wide-spectrum agent, which exhibits activity against several wound pathogens, including Gram-positive and -negative bacteria and fungi. Moreover, chlorhexidine washes have recently been found to reduce in vitro viability of polymicrobial wound biofilms and provide a unique proteomic response in skin models [5]. The amine groups present in the drug structure might also have a positive effect in the viability of mammalian cell lines, as suggested by several studies based on surface amination as a biofunctionalization process [6,7,8]. However, the therapeutic success of chlorhexidine-based treatments could be limited by skin irritation, intrinsic cytotoxic activity towards human fibroblasts and the presence of organic matter from wound debris, which reduce the antibacterial activity of the molecule [4]. These drawbacks could be overcome by designing drug–clay nanohybrids, in which drug molecules are adsorbed or encapsulated within the core of clay mineral structures. 

Clay minerals, among them bentonite (montmorillonite) and halloysite, are known in the pharmaceutical field as low-cost, naturally abundant and eco-friendly inorganic excipients. They also provide many advantages in tissue regeneration, such as hydrophilicity, good dispersibility, biocompatibility and drug incorporation capacity [9,10,11], being able to modulate the release of incorporated biomolecules and/or protect them from unfavorable environments. Furthermore, due to their surface electrostatic properties, clay minerals can act as an ideal platform to activate the coagulation cascade at the wound site, thus promoting hemostasis [12].

Given these premises, in this work the antiseptic chlorhexidine has been loaded into bentonite and halloysite structures to obtain nanotherapeutic materials aimed at healing chronic skin injuries. Both inorganic carriers have special crystalline arrangements, allowing them to entrap drug molecules, mainly via cation exchange (bentonite) or other mechanisms, such as hydrogen bonding and/or van der Waals interactions (bentonite and halloysite) [10,11]. Exchangeable cations are present in bentonite interlayers, which can be replaced by basic drug molecules, forming intercalate structures with expanded clay basal spacing [13]. Halloysite possess a hollow nanotubular morphology which allows spontaneous encapsulation of small drug molecules in the inner silicate lumens [14]. The obtained hybrids should provide higher contact area with target pathogens, (achieving greater therapeutic efficacy against resistant infections) [15], preserve drug effectiveness at the wound environment and reduce cytotoxicity.

Nanohybrids were prepared following a traditional solid–liquid intercalation technique. This methodology allows spontaneous interactions taking place between drug molecules and clay mineral particles in aqueous dispersions [13]. It is, however, a long process, which also requires separation and drying of the solid phases from the dispersion medium. Therefore, in order to optimize the preparation procedure, by reducing the time and number of steps involved, the feasibility of the spray-drying process was also assessed as an alternative preparation technique.

Nanohybrid structures were then characterized by X-ray powder diffraction. FTIR and thermal analyses and complementary molecular modeling were used to investigate drug–clay interactions by means of classical mechanics calculations. Ultra-high resolution transmission electron microscopy coupled with X-EDS (X-ray energy-dispersive spectroscopy) and elemental maps, as well as zeta potential measurements, were also performed to check the hybrid (organic/inorganic) character of the nanomaterials. Preliminary in vitro cytotoxicity on human fibroblasts and antimicrobial activity measurements against *Staphylococcus aureus* and *Pseudomonas aeruginosa* were also performed to assess biocompatibility and microbicidal efficacy of the obtained hybrids.

## 2. Materials and Methods

### 2.1. Materials

Clay minerals (nanoclay hydrophilic bentonite (BEN) and halloysite nanoclay (HAL)) and chlorhexidine dihydrochloride (CHX, Figure 1) (white powder with ≥ 98% purity, MW 578.37 g/mol) were purchased from Merck Life Science S.L.U. (Madrid, Spain). All other chemicals and solvents were high-quality analytical grade and used as procured.

### 2.2. Methods

#### 2.2.1. Preparation of Drug–Clay Nanohybrids

Nanohybrids were prepared by spontaneous intercalation solution and spray-drying techniques.

##### Intercalation Solution Technique

CHX aqueous solution was prepared at a concentration close to saturation (0.01% w/v) to maximize the contact between drug molecules and clay mineral particles. Then, 1% w/v clay mineral aqueous suspensions were prepared by dispersing BEN and HAL powders in purified water at 23,000 rpm for 15 min (Ultra-Turrax^®^ T25 S5, IKA^®^—Werke GmbH & Co. KG, Staufen, Germany). The particle size distribution of the obtained dispersions was assessed by light-scattering technology (Mastersizer 2000LF, Malvern Panalytical Ltd., Madrid, Spain), resulting in statistical diameters shown in Table 1.

CHX solution (5 mL) was then added to 5 mL of clay mineral dispersions and stirred at 220 rpm for 24 h at room temperature (Stuart^®^ SSL2 reciprocating shaker, Bibby Scientific Ltd., Stone, UK). This time was considered long enough to reach the adsorption equilibrium, based on the experience of previous works on the interaction between drug molecules and clay minerals [16,17,18,19]. Drug–clay dispersions were then centrifuged at 6000 rpm for 20 min (centrifuge Z 326 K, HERMLE Labortechnik GmbH, Wehingen, Germany) and the CHX residual concentration in supernatant was determined by UV spectroscopy at 255 nm (UV-Vis spectrophotometer lambda 35, Perkin Elmer, Madrid, Spain). The entrapment efficiency (EE) was calculated using Equation (1), resulting in 67 ± 5.1% for BEN and 94 ± 1.1% for HAL. No further increase of EE was observed at higher clay mineral concentrations.
(1)$$EE\;\left(\%\right)=\left(\frac{\mathrm{initial}\;\mathrm{concentration}\;\mathrm{of}\;\mathrm{CHX}\;-\mathrm{residual}\;\mathrm{concentration}\;\mathrm{of}\;\mathrm{CHX}}{\mathrm{initial}\;\mathrm{concentration}\;\mathrm{of}\;\mathrm{CHX}}\right)\times 100$$

At the end of the experiments, solid phases (hereinafter known as nanohybrids) were dried at room temperature in a desiccator until residual moisture was lost. Then, dried drug–clay nanohybrids were milled in an agate mortar and sieved to separate the <63 μm fraction, in order to obtain a homogeneous and reproducible particle size for further characterizations. 

Experiments were carried out in triplicate.

##### Spray Drying

Drug–clay aqueous dispersions, 100 mL in volume, at drug–clay *w*/*w* ratios used in the intercalation solution technique were spray dried using a Mini Spray Dryer (BÜCHI^®^ B-290; Massó Analítica S.A., El Prat de Llobregat, Barcelona, Spain) with a nozzle of 0.7 mm diameter aperture. The feed rate was 9 mL/min and the nozzle air pressure was 6 bar. The inlet temperature was set at 200 °C, resulting in an outlet temperature of 130 °C. The airflow rate and the aspirator were 439 L/h and 40 m^3^/h (100%), respectively. The efficiency of the process, determined from the amount of solid actually recovered versus the theoretical amount of solid to be obtained by atomization of the aqueous phase, was always ≥ 50%.

#### 2.2.2. Characterization of the Nanohybrids

##### Solid-State Characterization Techniques

X-ray powder diffraction (XRPD) was carried out using a D8 DISCOVER diffractometer (Bruker, Madrid, Spain) with Cu Kα radiation ((λ = 1.5406 Å), 50 kV, 1 mA, 3–36° 2θ exploration range and PILATUS3R 100K-A detector.

Fourier transform infrared spectroscopy (FTIR) spectra were recorded on a FTIR spectrophotometer (JASCO 6200, with software SPECTRA MANAGER v2 and with an attenuated total reflectance (ATR) accessory; Jasco, Easton, MD, USA). Measurements were carried out from 400 to 4000 cm^−1^ at 0.25 cm^−1^ resolution.

Thermogravimetric analysis (TGA) and differential scanning calorimetry (DSC) were carried out using a METTLER TOLEDO mod. TGA/DSC1 with FRS5 sensor and a microbalance (precision 0.1 μg) (Mettler-Toledo GMBH, Cornellà del Lobregat, Barcelona, Spain). Samples (20–40 mg) were heated in nitrogen atmosphere at 10 °C/min in 30–935 °C temperature range.

Ultra-High Resolution Transmission Electron Microscopy (UHR-TEM) was performed by means of an analytical electron microscope (Titan G2 60–300, FEI Company, Thermo Fisher Scientific, Waltham, MA, USA) with a SUPER-X silicon-drift windowless energy dispersive X-ray spectroscopy detector. X-ray chemical element maps were also collected. The samples were directly deposited onto copper grids (300 mesh coated by formvar/carbon film, Agar Scientific, Monterotondo, Italy).

##### Zeta Potential Measurements

Zeta (ζ) potential measurements were carried out by electrophoretic light scattering, using a Malvern Zetasizer NanoZS (Malvern Panalytical B.V., Madrid, Spain). Nanohybrids and pristine component powders were dispersed in distilled water (0.05 % *w*/*v*) and measurements were performed at 25 °C with voltage and number of runs chosen in automatic selection mode. Equilibrium time of 60 s was established before measuring. Each measurement was run in triplicate. 

One-way ANOVA (analysis of variance) and Bonferroni’s post hoc Scheffé test for statistical analysis of the experimental groups were performed on the obtained data, using GraphPad PRISM software, version 5.01 (GraphPad, San Diego, CA, USA). Differences were significant at *p* values less than 0.05. 

#### 2.2.3. Molecular Modeling Methodology and Models

##### Models

The CHX is protonated in water, forming a dication with two water-solvated chlorine counterions [20]. This structure was generated sketching by hand, using the Materials Studio package [21] and taking into account the experimental information.

A slide of HAL nanotube, with the stoichiometry Al_2_Si_2_O_5_(OH)_4_, was obtained from a previous work [22]. Subsequently, a periodical crystal structure of this nanotube was generated using periodic boundary conditions [18,23,24]. The unit cell of HAL has the formula Al_76_Si_76_O_190_(OH)_152_, with 646 atoms and an internal diameter of 27 Å. This structure is a proper model to study the drug–clay adsorption process. To carry out the adsorption of the drug, a 1 × 1 × 4 supercell of HAL was generated with the formula Al_304_Si_304_O_760_(OH)_608_ and with 2584 atoms. The protonated CHX with two water-solvated chlorine counterions was adsorbed on the internal and external surface of the HAL.

The unit cell of a BEN model was taken from a previous work [25]. For the adsorption calculations a 4 × 2 × 1 supercell of the BEN model was created with formula Na_6_(Al_24_Mg_8_)Si_64_O_160_(OH)_32_. This supercell was designed by considering the molecular dimension of the drug and periodic boundary conditions. The adsorption complexes were generated taking into account that the protonated CHX molecule (^2+^) is adsorbed on the BEN by cation exchange. For every molecule that enters, two sodium atoms leave the interlayer space. The adsorption of one and two CHX molecules in the interlayer space of 4 × 2 × 1-supercell montmorillonite was studied. In these complexes, 48 molecules of water per supercell were included in the interlayer space considering the water percentage measured experimentally.

##### Molecular Modeling Methodology

The geometry optimization of the protonated CHX was performed with the INTERFACE force field [26] based on empirical interatomic potentials and using the Forcite code [21].

On the other hand, the optimization of the unit cell of the HAL nanotube structure was performed with quantum mechanical calculations by using DFT with the CASTEP code [21]. The functionals used were GGA and PBE. On-the-fly-generated (OTFG) ultrasoft pseudopotentials were used with Koelling–Harmon relativistic treatment [27], and the cutoff energy of the calculation was 300 eV [21]. After the optimization of the HAL unit cell, the 1 × 1 × 4 supercell was created to study the adsorption of the drug. The water molecule was optimized using the same methodology as the CHX.

The drug was placed on the internal and external surface of the HAL using the Monte Carlo method and the optimized structure of the clay was fixed, except the hydrogen atoms of the internal surface. The geometry of the complexes was optimized. Subsequently, 100 ps of NVT (canonical ensemble, where the amount of substance (N), volume (V) and temperature (T) are conserved) molecular dynamic simulation was performed to equilibrate the systems using Forcite with INTERFACE [21,26]. These clay mineral structures have many secondary minimum-energy wells, and this ensemble showed a consistent exploration. Later, the drug–clay complexes were filled with previously optimized water molecules using a Monte Carlo method with INTERFACE [21], reaching a density of 1 g/cm^3^ (~2250 water molecules). The resultant adsorption complex was optimized with INTERFACE.

The adsorption energy of the complexes was estimated using the following equation:*E*_ads_ = (*E*_complex_) − (*E*_CHX_ + *E*_HAL_)(2)

For this, the energy of the optimized adsorption complex was calculated (*E*_complex_), as well as the energy of both components separately, namely, the drug and HAL with the same amount of water as the optimized complexes. After an optimization and dynamic equilibration (100 ps of NVT), the energies were calculated (*E*_CHX_ and *E*_HAL_), with the INTERFACE [21]. Note that the calculated adsorption energy includes the hydration energy of the drug.

In addition, the optimization of the BEN structure was performed with the INTERFACE force field using Forcite with a cut-off of 18.5 A. To study the adsorption of the drug, one and two molecules of protonated CHX were adsorbed using Monte Carlo method with the INTERFACE force field [21]. Different conformations and orientations between the drug, the clay, 48 water molecules and the sodium cations were randomly explored. The more stable drug–clay complexes were selected. Later, the geometry of the complex was optimized at variable volume, followed by 100 ps of NPT (isothermal–isobaric ensemble, where the amount of substance (N), pressure (P) and temperature (T) are conserved) molecular dynamic simulation to equilibrate the system using Forcite with INTERFACE [21]. Note that the number of water molecules was selected in agreement with the percentage of water measured in the complexes according to the TGA experimental data.

This adsorption energy of the complexes was estimated taking into account that the adsorption of the protonated CHX in the BEN interlayer is a cation exchange process. Furthermore, the cation–mineral surface and cation–water interactions are also considered for the calculation of adsorption energy and should remain similar before and after the cation exchange process. Then, we created and optimized a periodic box with two Na^+^ and two Cl^−^ ions solvated with 48 water molecules and another periodic box of CHX dichloride with two Cl^−-^ and 48 water molecules. The adsorption energy can be estimated by the following equations (3 and 4, for the adsorption of one or two drug molecules, respectively):*E*_ads_ = *E*_(BEN-6Na_^+^_-CHX_^2+^_-48w)_ + *E*_(2Na_^+^_-2Cl_^+^_-48w)_ − (*E*_(BEN-8Na_^+^_-48w)_ + *E*_(CHX_^2+^_-2Cl_^−^_-48w)_)(3)
*E*_ads_ = *E*_(BEN-4Na_^+^_-2CHX_^2+^_-48w)_ + *E*_(2Na_^+^_-2Cl_^+^_-48w)_ − (*E*_(BEN-6Na_^+^_-CHX_^2+^_-48w)_ + *E*_(CHX_^2+^_-2Cl_^−^_-48w)_)(4)
where *E*(_BEN--6Na_^+^_-CHX_^2+^_-48w_) is the energy of the BEN with six Na^+^ cations, one CHX^2+^ and 48 waters per 4 × 2 × 1 supercell; *E*(_2Na_^+^_-2Cl_^−^_-48w_) is the energy of the model with two Na^+^ and two Cl^−^ and 48 molecules of waters; *E*_(BEN-8Na_^+^_-48w)_ is the energy of the BEN with eight Na^+^ cations and 48 water molecules; *E*(_CHX_^2+^_-2Cl_^−^_-48w_) is the energy of the model of one CHX^2+^, two chlorine anions and 48 water molecules; and *E*(_BEN-4Na_^+^_-2CHX_^2+^_-48w_) is the energy of the adsorption complex model with the 4 × 2 × 1 BEN with four Na^+^ cations, two molecules of CHX^2+^ protonated drug and 48 water molecules.

#### 2.2.4. In Vitro Biocompatibility and Cell Migration Properties Measurements

The biocompatibility properties of nanohybrids have been assessed on normal human dermal fibroblasts (NHDF), from juvenile foreskin (PromoCell, WVR, Milan, Italy). NHDF (2nd–5th passages) were cultured in polystyrene flasks in complete medium (CM), namely Dulbecco’s Modified Eagle’s Medium (DMEM, Merck Life Science S.r.l., Milan, Italy), supplemented with 10% *v*/*v* heat-inactivated Foetal Bovine Serum (FBS) (VWR International S.r.l, Milan, Italy), and with 1% *v*/*v* antibiotic-antimycotic solution (Merck Life Science S.r.l., Milan, Italy). Cells were kept incubated at 37 °C in 5% CO_2_ atmosphere. Drug–clay nanohybrid powders were prepared in sterile conditions and subjected to UV-irradiation for 2 cycles of 20 min before their usage. After sterilization, 0.50 mg/mL suspensions of all the samples (nanohybrids obtained by intercalation technique and by spray drying) were prepared in CM. The same treatment and dilution were employed also for reference controls, namely pristine BEN, HAL and CHX. Then, 100 µL of cells (at a density of 20,000 cells/cm^2^) were seeded in a 96-well multiwell together with 100 µL of samples. Six replicates were performed for each sample; CM was used as reference. After 24 h of incubation at 37 °C in 5% CO_2_ atmosphere, an Alamar blue assay was performed. In detail, the medium was removed and 100 µL of a 10% *v*/*v* solution of Alamar blue in DMEM was added to each well and left in contact for 3 h. After 3 h, fluorescence was detected by means of a multi-mode microplate reader (FLUOstar Omega Microplate Reader, BMG LabTech, Ortenberg, G) at two different wavelengths: at 570 nm to detect the reduced form (red) of the Alamar Blue, and 655 nm to detect the oxidized one (blue).

As for cell migration assay, the gap closure assay was performed employing a Culture-Insert 2 Well pre-inserted into a 24-well multiwell (Ibidi, Giardini, Milan, Italy), which consists of a special insert consisting of two cell chambers (growth area: 0.22 cm^2^ each) divided by a septum (500 ± 100 μm in width), intended to simulate a cell-free gap. Fibroblasts were seeded in each chamber at 10^5^ cells/cm^2^ concentration. After 24 h of incubation at 37 °C in 5% CO_2_ atmosphere, cells reached a confluence state, and the insert was removed, allowing the obtainment of two areas of cell substrates divided by the cell-free gap. The cell substrates were put in contact with 500 μL of each sample. Spray-dried samples were treated in the same way as for the biocompatibility assay, and the concentration of usage was 0.25 mg/mL in CM for each test. At prefixed times t_0_ (0 h), t_1_ (24 h) and t_2_ (72 h), microphotographs were taken to evaluate cell migration in the gap.

#### 2.2.5. Time-Kill Studies

Time-kill kinetics assays were performed to check the antimicrobial activity of nanohybrids against the bacteria strains *Staphylococcus aureus* ATCC 6538 and *Pseudomonas aeruginosa* ATCC 15442. Both strains are chosen as model bacteria, because they are frequently found on the skin microbiota, being able to become opportunistic pathogens and causes of skin infections. Killing time was determined as the exposure time required to kill a standardized microbial inoculum. Bacteria used for killing-time evaluation were grown overnight in Tryptone Soya Broth (Oxoid; Basingstoke) at 37 °C. The bacteria cultures were centrifuged at 2000 rpm for 20 min to separate cells from broth and then suspended in saline phosphate buffer (PBS) (pH 7.3). The optical density of the microbial suspensions was adjusted to A = 0.3 (wavelength 650 nm), which corresponds to a bacterial titer of 1 × 10^7^–1 × 10^8^ CFU/mL (Colony Forming Units per milliliter). An exact amount of nanohybrids was added to the microorganism suspensions to obtain a nanohybrid concentration of 100 mg/mL. For each microorganism, a suspension was prepared in PBS and used as control. Bacterial suspensions were incubated at 37 °C. Viable microbial counts were evaluated after contact for 24, 48 and 72 h with the samples and microorganism suspensions grown in the same conditions and used as control. The bacterial colonies were enumerated in Tryptone Soya Agar (Oxoid; Basingstoke) after incubation at 37 °C for 24 h. The microbicidal effect value was calculated for each test organisms and contact times according to Equation (5) [28]:(5)$$ME=\log N_{c}-\log N_{d}$$
where *N_c_* is the number of CFU of the control microbial suspension and *N_d_* is the number of CFU of the microbial suspension in the presence of nanohybrids. 

## 3. Results and Discussion

### 3.1. Characterization of the Nanohybrids

#### 3.1.1. Solid-State Characterizations

X-ray powder diffractograms (XRPD) of nanohybrids and pristine components are plotted in Figure 2a,b for BEN- and HAL-based materials, respectively. CHX exhibits a crystalline pattern with main peaks at about 8.5°, 12.7°, 13.4°, 15.9°, 20.9°, 23.7°, 26.6° and 34.4° 2θ, which are consistent with those found in the literature [29,30]. BEN shows the d_001_ basal reflection peak at 6.7° 2θ, approximately. According to Bragg’s law, this value gives an interlayer spacing of 13.15 Å, which is ascribable to a smectite saturated with a mixture of monovalent–divalent exchangeable cations [31]. For both bentonite-based nanohybrids, d_001_ basal reflection shifts to lower 2θ values (around 5.88 2θ and 6.22 2θ, for CHX–BEN and CHX–BEN SD (spray dried), respectively) in comparison with the pristine clay mineral. These results indicate that d_001_ basal spacing has been expanded to ≅15 Å (CHX–BEN) and ≅14 Å (CHX–BEN SD), suggesting that both methods of preparation were able to produce the effective intercalation of CHX molecules into the clay mineral structure. HAL shows d_001_-basal reflection at 12° 2θ (Figure 2b), corresponding to a basal spacing of 7 Å which is typical of the dehydrated form of the clay mineral [18]. No expansion in basal spacing has been observed in the nanohybrids (CHX–HAL and CHX–HAL SD). These results are coherent with the non-swelling nature of this clay mineral. Therefore, the presence of the drug in the hybrids needs to be confirmed by further characterization.

Moreover, no crystals of CHX were observed, indicating that all CHX has become amorphous in the nanohybrids in both minerals.

In Figure 3 FTIR spectra of nanohybrids in comparison with pristine components are given. Pristine BEN, HAL and CHX show the main vibrational bands described in the literature [18,19,32]. The hybrid character of both CHX–BEN and CHX–BEN SD is confirmed by the presence of main bands of BEN, as well as bands of the CHX molecule in the region 3300–3100 cm^−1^. Moreover, the absence of the N–H stretch band of the secondary amine salt of CHX around 2956 cm^−1^ could confirm that, in both nanohybrids, drug molecules have been intercalated into the clay mineral via cation exchange, as pointed out by the basal spacing expansion in XRPD patterns. In the case of HAL-based nanohybrids, the intense vibrations of the clay mineral obscure the CHX bands in practically the entire range of the spectrum. Absence of vibrational bands of the drug in the region 3300–3200 cm^−1^, especially in the case of CHX–HAL, could indicate the inclusion of the drug molecules into the clayey nanotubes. Nevertheless, further analyses are needed to confirm the identity of the drug in the HAL-based nanohybrids.

TGA and DSC curves of the samples are given in Figure 4 and Figure 5, respectively. The TGA curve of CHX shows an abrupt loss of mass above 240 °C, leading to a residual mass of 2% (*w*/*w*) at the end of the heating (935 °C). In the same temperature interval, the DSC curve shows an endothermic peak followed by an exothermic one. These are associated with melting and degradation of the drug [30]. A thermogram of BEN (Figure 4a) shows an initial mass loss of about 13% in the interval between 35 °C and 200 °C, approximately. The DSC curve exhibits, in the same range of temperatures, an endothermic phenomenon which can be ascribed to sample dehydration (loss of residual humidity and interlayer water molecules). At higher temperatures (above 600 °C) the TGA curve of BEN shows another loss of mass, leading to a total loss of 18% (*w*/*w*) of the initial amount of the sample.

This loss is associated with a wide endothermic band, in the DSC curve, due to dehydroxylation of the aluminosilicate [31]. BEN–nanohybrids (CHX–BEN and CHX–BEN SD) exhibit at the explored temperatures thermal behaviors (water loss and dehydroxylation) similar to those observed in the clay mineral (Figure 4a and Figure 5a). Drug melting peaks are not evident in DSC curves, corroborating the amorphization of CHX. The actual hybrid character of the samples can be justified by the difference in mass loss compared with pristine BEN, calculated according to the rule of mixtures for the residual masses at 935 °C and taking into account the water content of the samples [33], resulting in 0.66% (*w*/*w*) of drug. This result is consistent with the calculated encapsulation efficiency in intercalation/solution studies. The HAL thermogravimetric curve exhibits an initial mass loss around 100 °C and a second loss in the interval 400 °C–600 °C. Both steps are associated with endothermic phenomena in the DSC profile and can be attributed to dehydration of the physically adsorbed water and to dehydroxylation of the structural aluminol groups [18]. The presence of the drug in the hybrids is also evidenced by the difference in the masses lost compared to the pristine clay, the amount being in line with that expected from intercalation/solution technique (0.83% *w*/*w*).

Figure 6 shows microphotographs and EDX spectra of pristine components. Results of complete UHR-TEM characterization of nanohybrids and SD nanohybrids are given in Figure 7 and Figure 8, respectively. A microphotograph of CHX (Figure 6A) shows aggregates of irregular quadrangular particles. The EDX spectrum, performed in the marked zone (red square), reveals the presence of N and Cl as characteristic elements of this drug molecule. BEN (Figure 6B) shows the typical morphology of montmorillonite, consisting of lamellar aggregates. The EDX analysis confirms that it is an aluminum–magnesium silicate with Na and Ca as main exchangeable cations, as observed by XRPD. CHX–BEN (Figure 7A) exhibits a general morphology similar to that of the clay mineral. The presence of the drug is confirmed by means of the EDX analysis. The elemental composition, performed in three different areas, evidences the characteristic components of BEN (Si, O, Al, Mg, Na and Ca) and those of the organic molecule with N and Cl as main CHX indicators. 

The organic/inorganic nature of CHX–BEN can be corroborated in Figure 7B, where the elemental maps of the sample not only exhibit the presence of characteristic components of the drug and clay mineral (Cl and Si, respectively) but also a homogeneous distribution of the drug in the whole sample. 

UHR-TEM microphotographs of HAL and CHX–HAL reveal the characteristic hollow tubular morphology of the clay mineral both in native HAL (Figure 6C) and the hybrid system (Figure 7C). EDX spectra of selected areas are also included in both cases, showing peaks of the inorganic nanotubes (O, Al and Si), as well as the presence of Cl and N, confirming the effective loading of the drug. X-ray maps (Figure 7D) evidence the elemental composition of the nanotubes in the nanohybrids, as well as the homogeneous distribution of CHX in the hybrid systems.

As can be expected from the SD preparation technique, both SD nanohybrids appear as spheroidal aggregates, in which layered (Figure 8A) or tubular (Figure 8C) particles can be recognized, depending on the clay component. Both samples also reveal a uniform distribution of the drug in the clayey structures (Figure 8B,D for CHX–BEN SD and CHX–HAL SD, respectively). 

#### 3.1.2. Zeta Potential Measurements

Results of ζ-potential measurements (Figure 9) show that, as it was expected, pristine clay particles have negative surface charges, with ζ-potential values ranging from about −23 mV (BEN) to −14 mV (HAL). These values are in line with those described in the literature for smectite and kaolin mineral groups [34].

In the case of CHX, which is in the form of the dihydrochloride salt, positive ζ-potential is observed, reaching values of about 30 mV (29.8 ± 1.93 mV). All nanohybrids exhibit intermediate ζ-potential values between the drug and the corresponding clay mineral. These results indicate that drug–clay electrostatic interactions are involved in the formation of the hybrids, leading to a partial neutralization of negative potential of the clays. In particular, ζ-potentials increase from −23 mV (BEN) to −20 ± 1.44 mV (CHX–BEN) and −21 ± 0.71 mV (CHX–BEN SD) for BEN-based nanohybrids, and from −14 mV (HAL) to −9.6 ± 0.17 mV (CHX–HAL) and −8.8 ± 0.36 mV (CHX–HAL SD) for HAL-based ones. It can be observed that neutralization is stronger in the case of halloysite nanohybrids, which show increase in ζ-potential of up to five units with respect to pure clay, while in bentonite-based nanohybrids the increase is approximately three units. This result suggests that electrostatic attraction between the drug molecules and the negatively charged edges of BEN may also take place, although to a lesser extent than in the HAL. This superficial interaction between CHX molecules and BEN would add to the main mechanism of cation exchange observed by XRPD. Moreover, the extent of neutralization is significantly higher (*p* < 0.05) in the case of the CHX–HAL SD nanohybrid. It can be suggested that drug–clay intermolecular interactions were amplified by the high temperatures provided by the spray-drying process.

### 3.2. Molecular Modeling

#### 3.2.1. CHX Adsorption on the Internal and External Surfaces of the Halloysite

Initially, we performed an investigation on the adsorption of the CHX on the internal and external surface of the HAL by theoretical calculations. For that, the optimized CHX was placed in the DFT-optimized HAL which has the structure fixed, except the hydrogen atoms of the internal surface. The geometry optimization of this structure was carried out using INTERFACE, followed by a NVT molecular dynamic simulation to equilibrate the model. Subsequently, the structure was filled with water molecules, and the resultant adsorption complex was optimized with INTERFACE (Figure 10).

In the adsorption complex with the drug adsorbed on the internal surface of the HAL (Figure 10a,b), the CHX is positioned with an orientation parallel to the inner surface of the nanotube. The aromatic rings are mainly parallel to the surface. Hydrogen bonds between the CHX and HAL were found between the N atoms of the drug and the hydrogen atoms of the internal surface of the HAL with d (N…HOAl) around 1.9 Å. In addition, the chlorine anions have interactions with the amino H atoms of the CHX with d (NH_2_…Cl) around 1.9 Å. At the same distance, these ions also interact with the hydrogens of the water. On the other hand, when the drug is adsorbed on the external surface of the HAL (Figure 10c,d), the hydrogens of the amino groups of the drug were interacting with the oxygens of the external surface of the HAL with d (NH_2_…OSi) around 2.5 Å. In the same way as in the previous system, the chlorine anions had interactions at 1.9 Å with the amino H atoms of the drug and with the hydrogen atoms of the water.

Lastly, the adsorption energies of both models were compared according to the Equation (2) *E*_ads_= (*E*_complex_) − (*E*_CHX_ + *E*_HAL_). The results showed that both adsorption models presented a negative adsorption energy. Therefore, the adsorption of the drug is favorable on the internal and external surfaces of the HAL nanotube. However, the CHX adsorption is more favorable on the external surface (−168.3 kcal/mol) than on the internal surface (−59.7 kcal/mol).

#### 3.2.2. CHX Adsorption in the Interlayer Space of the Bentonite

The adsorption of the drug in the interlayer space of the BEN was studied. For that, the optimized CHX, the water and the sodium cations were placed in the interlayer space of the clay using the Monte Carlo method. Subsequently, we carried out the geometry optimization of this structure followed by an NPT molecular dynamic simulation to equilibrate the model using INTERFACE. Figure 11 shows the resultant adsorption complexes.

In both adsorption complexes (Figure 11), the aromatic rings of the CHX are positioned with an orientation perpendicular to the interlayer surface of the BEN. Interactions between the CHX and BEN were found, mainly between the hydrogens of the nitrogen group of the drug and the oxygens of the surface with d (NH_2_…OSi) around 2.0 Å.

In addition, the interlayer spaces were calculated. Firstly, the interlayer space of the BEN supercell with 8 sodium cations and 48 waters proved to be *d* (001) = 13.22 Å, reproducing the exact experimental spacing (13.18 Å). The complex with one molecule of drug showed an interlayer space of *d* (001) = 15.48 Å. This result is in agreement with the experimental result shown by the X-ray interaction product. However, the interlayer space of the complex with two CHX molecules was 17.10 Å, higher than that measured experimentally. This indicates that, in the interaction product, only one drug molecule is adsorbed in the interlayer space of the BEN supercell in a monolayer disposition. The rest of the drug is adsorbed on the external surface of the clay.

On the other hand, the adsorption energies of both complexes were estimated according to the Equations (3) and (4). The results showed that the adsorption model with one molecule of drug presented a negative adsorption energy (−179.56 kcal/mol). However, the adsorption complex with two CHX molecules showed positive adsorption energy (17.75 kcal/mol). Therefore, only the adsorption of one molecule of drug is favorable on the interlayer space of the BEN supercell. This corroborates the interlayer spacing data which also show the adsorption of one CHX molecule per 4 × 2 × 1 supercell.

### 3.3. In Vitro Biocompatibility and Cell Migration Properties Measurements

The cytotoxic effect of CHX–BEN, CHX–BEN SD, CHX–HAL and CHX–HAL SD was assessed on NHDF. The sample concentration used, namely 0.50 mg/mL, was selected on the basis of a preliminary biocompatibility assay comparing the effect on NHDF viability of three different concentrations of interaction products (1 mg/mL, 0.75 mg/mL and 0.5 mg/mL); 0.5 mg/mL proved to be the concentration that achieved biocompatibility for all samples (data not shown). NHDF cells represent one of the most suitable model cell lines for in vitro evaluations of scaffolds intended for the treatment of skin injuries, as they are the most abundant cell type in all connective tissues and play an important role in wound healing process [35,36].

Results of the cytotoxicity test are reported in Figure 12a as the percentage of living cells after contact with the samples for 24 h. As can be observed, all the samples have a good biocompatibility with NHDF cells, as the values of cell viability percentage exceeded 80%. In particular, the CHX–HAL and CHX–HAL SD samples proved to be statistically different from the references HAL and CHX and, moreover, no statistical difference was observed when such samples were compared with CM. As for CHX–BEN SD higher cell viability percentage values with respect to BEN were observed and, furthermore, it displayed no statistical difference with CM. Indeed, no statistical difference was observed in the case of nanohybrid CHX–BEN compared to the pristine components, and lower values of cell viability % were observed in comparison with CM. Based on the biocompatibility assay, SD samples were selected for cell migration test. The better performance of HAL-based spray-dried nanohybrids was observed also during cell migration test, as can be observed in Figure 12b. In fact, CHX–HAL SD was able to promote cell migration/proliferation in the gap during the 72 h test and, in particular, a complete gap closure was observed. Therefore, this nanohybrid showed effective capability of in vitro wound healing promotion. However, the BEN-based nanohybrid was not able to promote cell migration from the side towards the cell-free zone (Figure 12b).

### 3.4. Time-Kill Studies

Time-kill studies provide basic information on the antimicrobial effects of nanohybrids as a function of time, taking into account that an effective control of the bacterial infections into the wound bed may promote wound healing. Microbicidal effects (ME) vs. time of nanohybrids against *Staphylococcus aureus* and *Pseudomonas aeruginosa* are given in Figure 13a,b, respectively. All nanohybrids show higher ME against *S. aureus* than against *P. aeruginosa* and the highest levels are observed in the case of CHX–HAL SD. Moreover, for both bacterial strains, SD nanohybrids exhibit higher antimicrobial activities than those obtained by intercalation solution techniques. Once more, as hypothesized by cytotoxicity studies, a larger surface area could increase antimicrobial activity. These results could agree with other studies, showing enhanced antimicrobial activity in drug–clay nanocomposites [37]. Moreover, except for CHX–HAL against both strains and CHX–BEN against *P. aeruginosa*, the antibacterial effect persists for 48 h on contact, being a very promising result in the application of these nanohybrids against chronic wound infections.

## 4. Conclusions

Nanohybrids based on CHX–BEN and CHX–HAL were effectively prepared via a spontaneous intercalation–solution technique. The spray-drying process was also able to effectively entrap CHX molecules into the clay mineral structures, also having the advantage of being a quick process that takes place in a single stage. Theoretical calculations confirmed the encapsulation of CHX into HAL, with hydrogen bonds taking place between the amino groups of the drug and the internal and external surfaces of HAL. Moreover, they confirmed the favorable adsorption of one molecule of CHX in the interlayer space of the BEN supercell in a monolayer disposition according to the experimental data. Nanohybrids, especially those obtained by the spray-drying procedure, exhibited good biocompatibility toward human fibroblast cultures and persistent microbicide activity against Gram-negative (*P. aeruginosa*) and Gram-positive (*S. aureus*) bacteria. They were able to prevent the bacterial contamination of wounds. CHX–HAL SD was also able to effectively promote proliferation of fibroblast cultures in in vitro wound healing assays. In conclusion, the obtained nanohybrids can be considered promising tools to enhance wound healing in infected wounds. Future studies will be performed in order to incorporate these materials into biopolymeric substrates able to mimic physiological characteristics of skin tissue, providing multifunctional advanced nanotherapeutic systems.

## Figures and Tables

**Figure 1 pharmaceutics-15-01140-f001:**
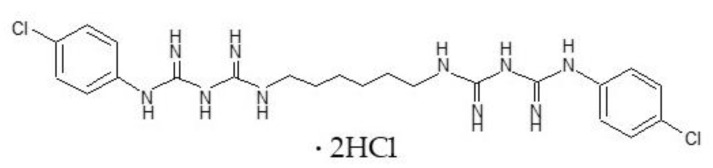
Chemical structure of CHX.

**Figure 2 pharmaceutics-15-01140-f002:**
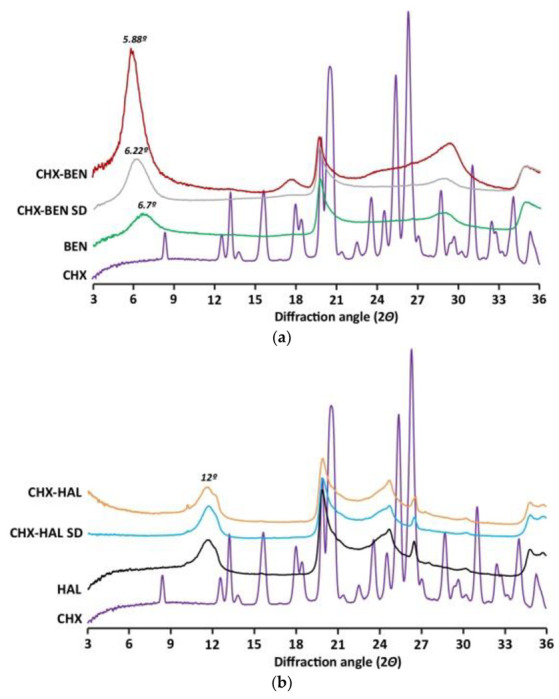
XRPD patterns of nanohybrids and pristine components: (**a**) BEN-based nanohybrids; (**b**) HAL-based nanohybrids. CHX: chlorhexidine dihydrochloride; BEN: bentonite; HAL: halloysite; CHX–BEN and CHX–HAL: nanohybrids obtained by intercalation solution technique; CHX–BEN SD and CHX–HAL SD: nanohybrids obtained by spray-drying technique.

**Figure 3 pharmaceutics-15-01140-f003:**
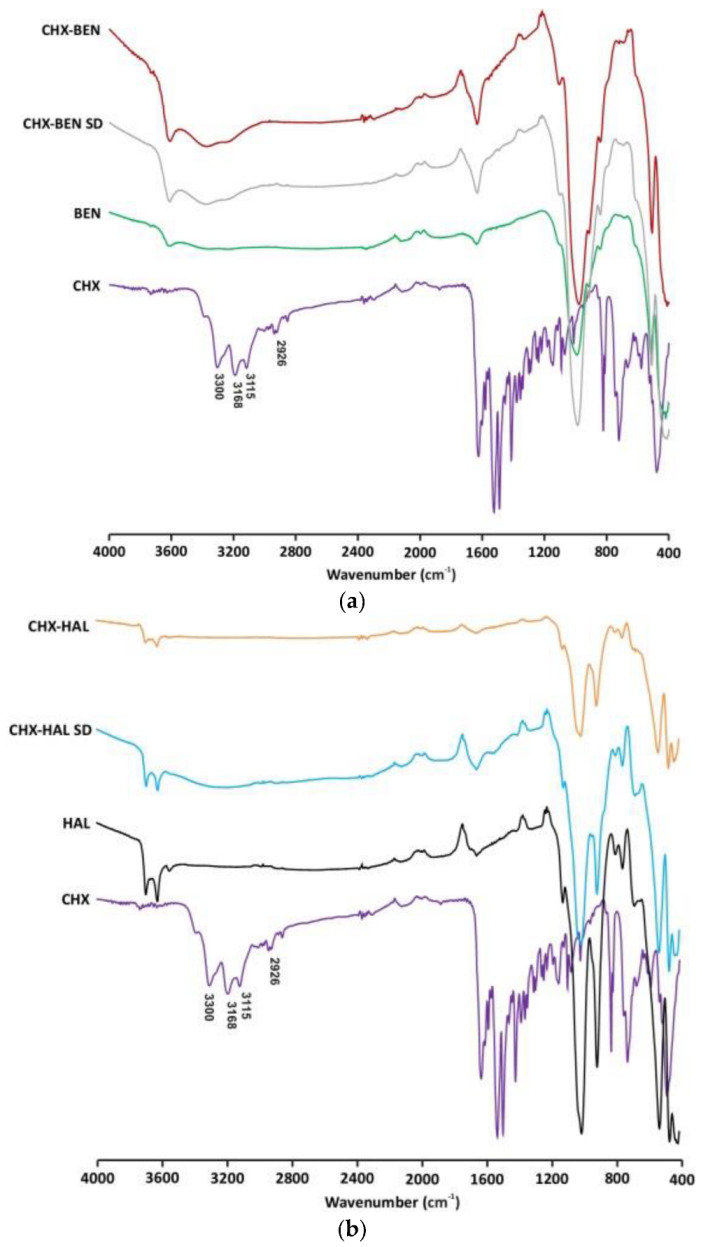
FTIR spectra of nanohybrids and pristine components: (**a**) BEN-–based nanohybrids; (**b**) HAL-–based nanohybrids. CHX: chlorhexidine dihydrochloride; BEN: bentonite; HAL: halloysite; CHX–BEN and CHX–HAL: nanohybrids obtained by intercalation solution technique; CHX–BEN SD and CHX–HAL SD: nanohybrids obtained by spray-drying technique.

**Figure 4 pharmaceutics-15-01140-f004:**
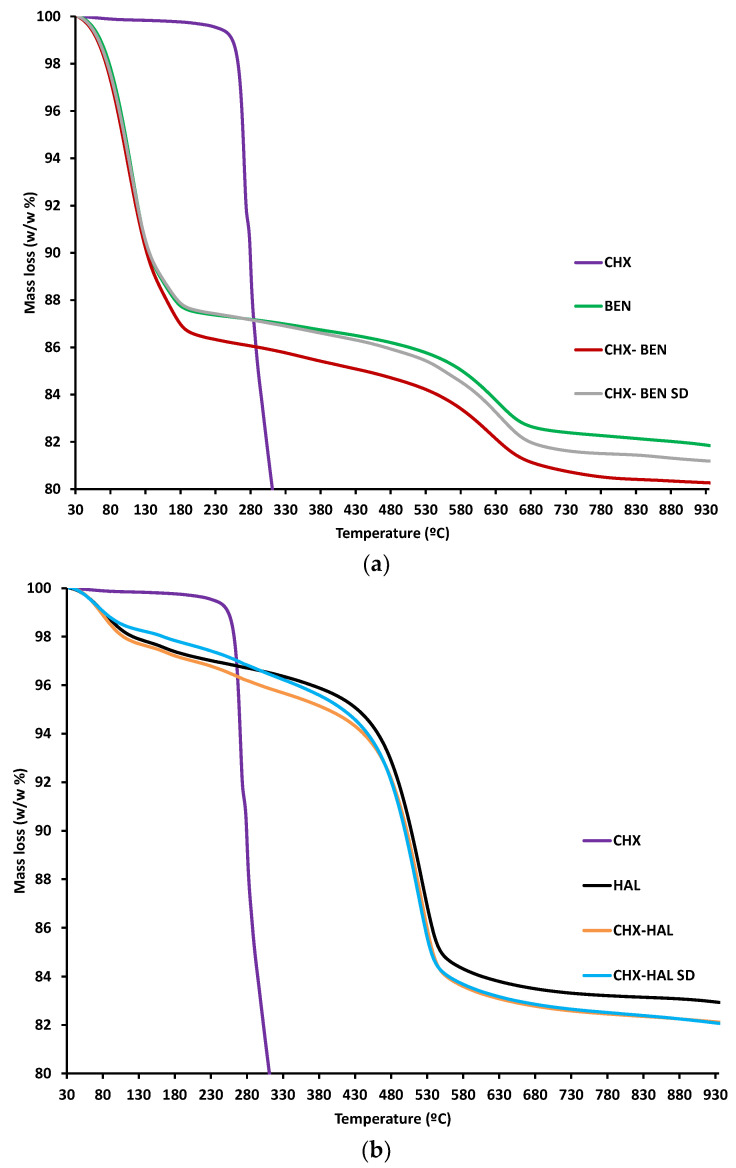
TGA curves of nanohybrids and pristine components: (**a**) BEN-based nanohybrids; (**b**) HAL-based nanohybrids. CHX: chlorhexidine dihydrochloride; BEN: bentonite; HAL: halloysite; CHX–BEN and CHX–HAL: nanohybrids obtained by intercalation solution technique; CHX–BEN SD and CHX–HAL SD: nanohybrids obtained by spray-drying technique.

**Figure 5 pharmaceutics-15-01140-f005:**
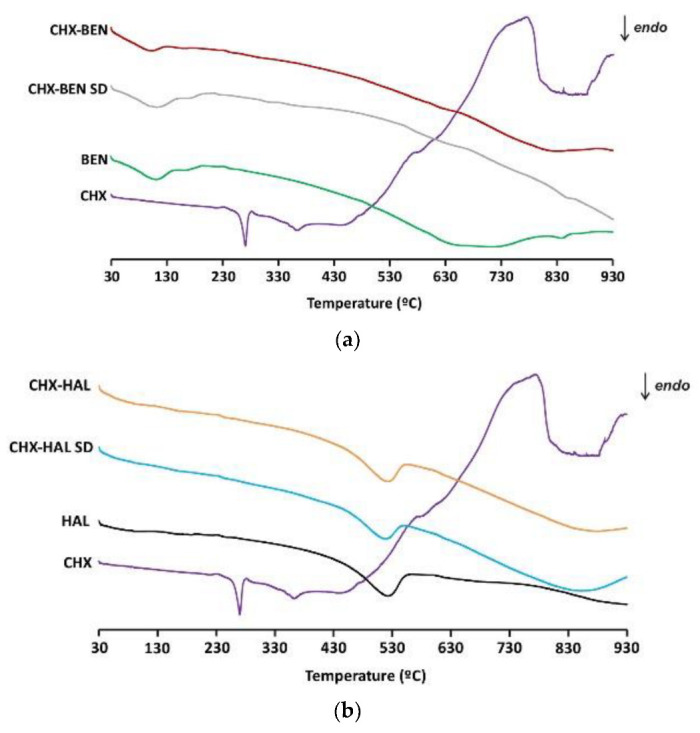
DSC curves of nanohybrids and pristine components: (**a**) BEN-based nanohybrids; (**b**) HAL-based nanohybrids. CHX: chlorhexidine dihydrochloride; BEN: bentonite; HAL: halloysite; CHX–BEN and CHX–HAL: nanohybrids obtained by intercalation solution technique; CHX–BEN SD and CHX–HAL SD: nanohybrids obtained by spray-drying technique.

**Figure 6 pharmaceutics-15-01140-f006:**
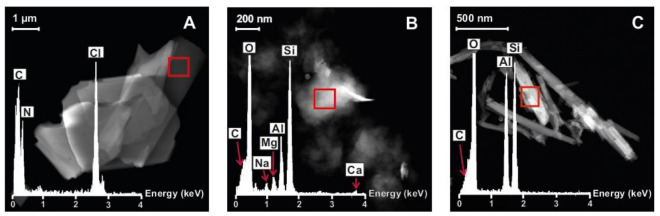
UHR-TEM microphotographs and EDX spectra of pristine components: CHX (**A**), BEN (**B**) and HAL (**C**).

**Figure 7 pharmaceutics-15-01140-f007:**
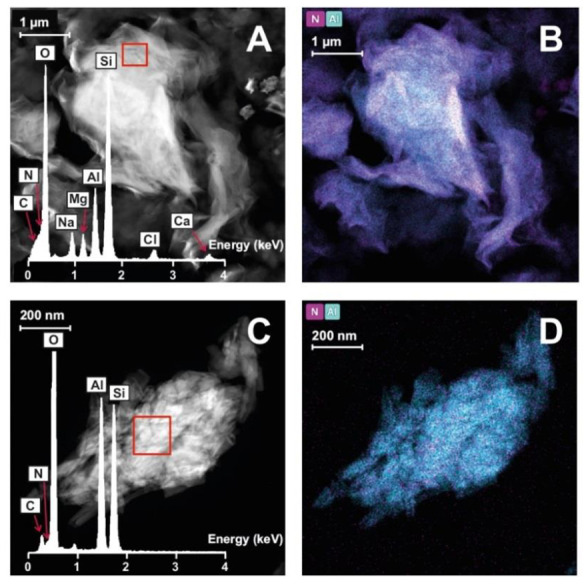
UHR-TEM microphotographs with EDX spectra and elemental maps of CHX–BEN (**A** and **B**, respectively) and CHX–HAL (**C** and **D**, respectively).

**Figure 8 pharmaceutics-15-01140-f008:**
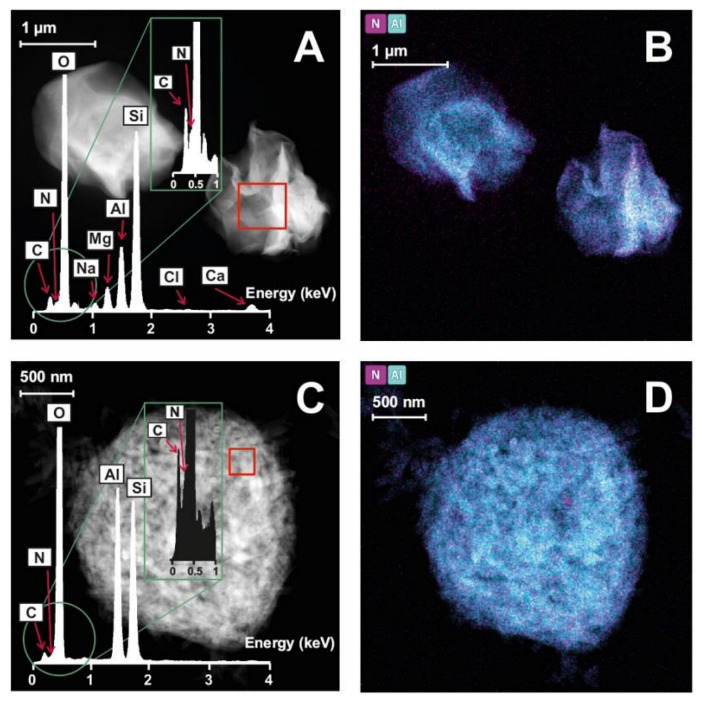
UHR-TEM microphotographs with EDX spectra and elemental maps of CHX–BEN SD (**A** and **B**, respectively) and CHX–HAL SD (**C** and **D**, respectively).

**Figure 9 pharmaceutics-15-01140-f009:**
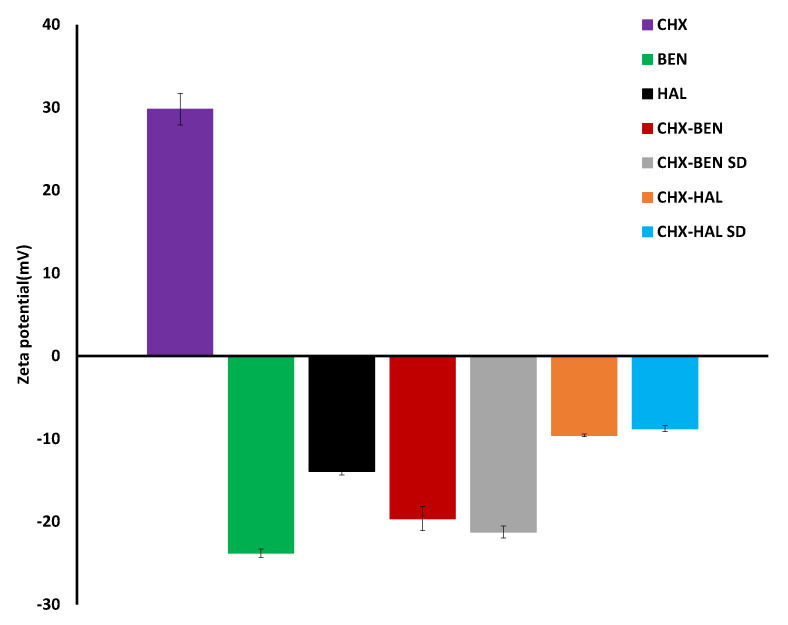
Values of ζ-–potential of nanohybrids and pristine components (mean values ± s.d.; *n* = 3). CHX: chlorhexidine dihydrochloride; BEN: bentonite; HAL: halloysite; CHX–BEN and CHX–HAL: nanohybrids obtained by intercalation solution technique; CHX–BEN SD and CHX–HAL SD: nanohybrids obtained by spray-drying technique.

**Figure 10 pharmaceutics-15-01140-f010:**
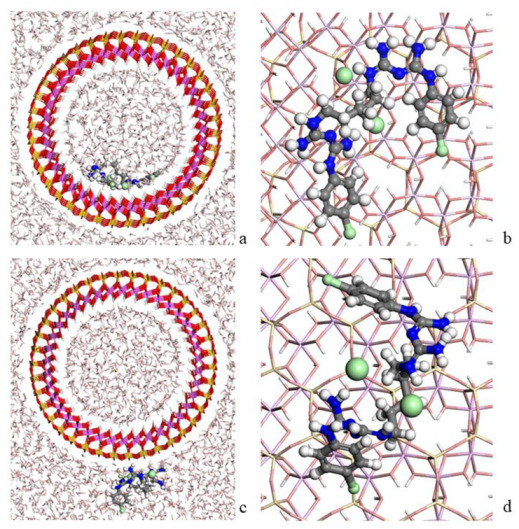
Complex of the CHX drug adsorbed on the internal (**a**,**b**) and external (**c**,**d**) surface of HAL nanotube from different views and with (**a**,**c**) and without (**b**,**d**) water solution. The atoms of silicon, aluminum, oxygen, hydrogen, carbon, nitrogen and chlorine are presented in yellow, pink, red, white, grey, blue and light green, respectively.

**Figure 11 pharmaceutics-15-01140-f011:**
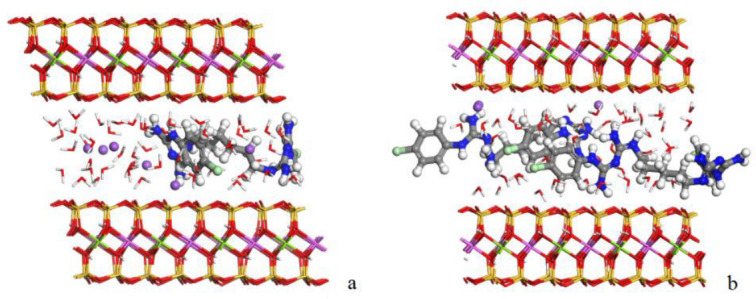
Complexes with one (**a**) and two (**b**) molecules of CHX drug adsorbed in the interlayer space of the BEN. The atoms of silicon, aluminum, magnesium, oxygen, hydrogen, carbon, nitrogen, chlorine and sodium are presented in yellow, pink, green, red, white, grey, blue, light green and purple, respectively.

**Figure 12 pharmaceutics-15-01140-f012:**
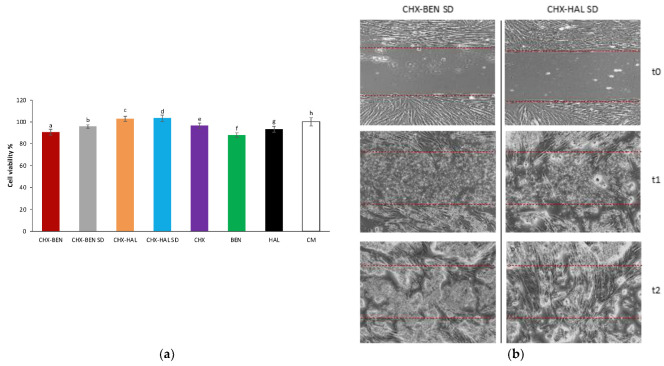
(**a**) Viability percentage values calculated after cell contact with samples CHX–BEN, CHX–BEN SD, CHX–HAL, CHX–HAL SD and references BEN, HAL and CHX (mean values ± s.d.; *n* = 6). CM was considered as control. ANOVA one-way, post hoc Scheffè test (*p* < 0.05: a vs. c, d, h; b vs. c, d, h; c vs. e, f, g; d vs. e, f, g; e vs. f; f vs. g, h; g vs. h. (**b**) Microphotographs (maginification 10X) of cell substrates taken at t_0_ (namely, after insert removal), at t_1_ (after 24 h of sample treatment) and at t_2_ (after 72 h of sample treatment), using optical microscope. Two red dotted lines were drawn corresponding to the cell-free gap so as to better visualize the area of cell migration. CHX: chlorhexidine dihydrochloride; BEN: bentonite; HAL: halloysite; CHX–BEN and CHX–HAL: nanohybrids obtained by intercalation solution technique; CHX–BEN SD and CHX–HAL SD: nanohybrids obtained by spray-drying technique.

**Figure 13 pharmaceutics-15-01140-f013:**
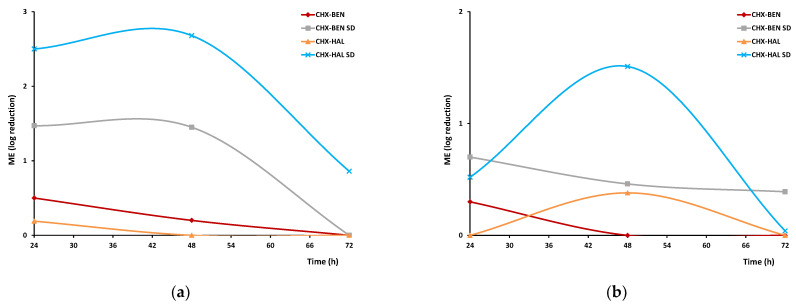
ME effect of nanohybrids against *Staphylococcus aureus* (**a**) and *Pseudomonas aeruginosa* (**b**). CHX: chlorhexidine dihydrochloride; BEN: bentonite; HAL: halloysite; CHX–BEN and CHX–HAL: nanohybrids obtained by intercalation solution technique; CHX–BEN SD and CHX–HAL SD: nanohybrids obtained by spray-drying technique.

**Table 1 pharmaceutics-15-01140-t001:** Statistical diameters of BEN and HAL aqueous dispersions (mean values ± s.d.; *n* = 3).

	d_10_ (μm)	d_50_ (μm)	d_90_ (μm)
BEN	0.969 ± 0.0078	2.890 ± 0.0516	9.369 ± 0.2524
HAL	1.495 ± 0.0163	5.175 ± 0.0304	14.663 ± 0.0962

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
