# Peer review of "Characterization and Molecular Modelling of Non-Antibiotic Nanohybrids for Wound Healing Purposes"

_pharmaceutics, 2023, doi:10.3390/pharmaceutics15041140_

Round 1

Reviewer 1 Report

Review of pharmaceutics-2276157

This is a very interesting manuscript, about the incorporation of chlorhexidine (CHX, a common antiseptic, but non-antibiotic) and clay (bentonite, halloysite) as nanohybrids for wound healing applications. The nanohybrids are projected to have antibacterial properties, while being non-lethal to mammalian cells. The characterizations are well performed. In addition, quantum mechanics is applied to confirm the performance of nanohybrids in molecular level. This paper will be a nice contribution not just to the pharmaceutical science, but also biomaterials science, nanotechnology, quantum chemistry, etc. There are several issues to be corrected in order to improve this manuscript, as follows:

1.       Please add the chemical structure of CHX in the manuscript.

2.       In the chemical structure of CHX, there are ten amine groups (compared to only two benzylchloride groups), that are highly likely become the supporter for the mammalian cell viability. This phenomenon of the positive effect of amine-rich surfaces towards mammalian cells is highly suggested to be incorporated in the Section 1 – Introduction, by referring to several literatures of surface amination as a biofunctionalization process such as:

  • Applied Surface Science 473 (2019) 838-847 https://doi.org/10.1016/j.apsusc.2018.12.216
  • Journal of Polymer Science Part B: Polymer Physics 51 (2013) 1361-1367 https://doi.org/10.1002/polb.23341
  • Biochemical Engineering Journal 78 (2013) 198-204 https://doi.org/10.1016/j.bej.2013.02.022

3.       Figure 12 vs Figure 11 vs Section 3.3 line 582: Please justify why the duration of the tests are not uniform. In Figure 12, the microbiocidal effect was tested for 72 h (peaked at 48 h), while in Figure 11 and line 582, the mammalian cell viability was tested only for 24 h. It is strongly suggested to add the data of cell viability test for 72 h.

4.       Line 234: Please add the definition of NVT (canonical ensemble, where the amount of substance (N), volume (V), and temperature (T) are conserved).

5.       Line 243: Please add the definition of NPT (isothermal-isobaric ensemble, where the amount of substance (N), pressure (P), and temperature (T) are conserved).

6.       Relate to NVT and NPT, why NVE (microcanonical ensemble, where the amount of substance (N), volume (V), and energy (E) are conserved) is not employed in this study? Please kindly add the explanation in the manuscript.

7.       Line 125: Please do not start a sentence with numbers. Suggestion for revision: CHX solution (5 mL) were then…

8.       Line 211-213: Please merge this short paragraph (one sentence only) with the next paragraph.

9.       Line 268: 2nd-5th --> superscripted nd and th

10.   Line 271: I believe it should be “MeRCK” instead of “MeRK”. But please confirm it.

11.   Line 318: Please check the decimal sign. Is it “13 point 15 Å” or “13 comma 15” Å?

12.   Reference 24: Please use “6th”, instead of “sixth”

Reviewer 2 Report

We congratulate the authors for the study and for choosing this research topic of major interest for the medical-pharmaceutical field in the context of the development of antibiotic resistance in skin infections, and not only.

We agree with the publication of the manuscript in Pharmaceutics Journal, after making the following revisions as follows:

-  in Intercalation solution technique subsection, explain and complete why dried drug/clay nanohybrids were sieved and separated for < 63 µm fraction (line 139);

- in Spray Drying subsection, explain and complete how the efficiency of the processes was determined, declared as more than 50% (lines 150-151).

- correct the orthography for: biocompatibility (line 40); socio-economic (52-53); unfavorable (73-74); hemostasis (76); cytotoxicity (103); titer (297); studies (380); to equilibrate (512; 543-544);

- line 571 – In vitro (no italics) 

Reviewer 3 Report

The manuscript from Valentino and collaborators describes the development and characterization of nanohybrids based on chlorhexidine dihydrochloride and clay minerals aiming for the applications in the treatment of chronic wounds.

The authors have rightly designed the research and the manuscript is well written and presented. However, some adjustments are needed before the final decision could be taken.

Major concerns 

- The authors use in lines 31, 572, 581 the term ‘proliferation’. However, they only assessed cell viability with Alamar blue assay. This conceptual mistake needs to be fixed.

- In the title and throughout the text, the authors claims that the material is developed for wound healing purposes. However, they did not performed any direct assay to show this property. I suggest that the authors perform  the simple and inexpensive scratch-wound assay. This would provided data about the ability of the proposed material on cell migration and proliferation.

Few adjustments:

- Line 88-89: Please consider to merge this sentence with other paragraphs to make the paragraph size more uniform.

- Please provide the mean of each abbreviation in the figure legends.

- Please make uniform the use of units as liter which is represented as “L” and “l”.

- Please attempt to the use of italics for latim words such as in vitro (for instance in line 578).

Round 2

Reviewer 1 Report

Review of pharmaceutics-2276157-v2

The manuscript has been improved significantly. It can be accepted in the present form. Thank you.

Reviewer 3 Report

The authors have improved the manuscript following the suggestions of the reviewers.